# Influence of Upcycled Post-Treatment Bark Biomass Addition to the Binder on Produced Plywood Properties

Aleksandra Jeżo [1], Anita Wronka [2,*], Aleksander Dębiński [3], Lubos Kristak [4], Roman Reh [4], Janis Rizhikovs [5] and Grzegorz Kowaluk [2]

1    Faculty of Wood Technology, Warsaw University of Life Sciences—SGGW, Nowoursynowska St. 159, 02-787 Warsaw, Poland
2    Institute of Wood Sciences and Furniture, Warsaw University of Life Sciences—SGGW, Nowoursynowska St. 159, 02-776 Warsaw, Poland
3    Institute of Forest Sciences, Warsaw University of Life Sciences—SGGW, Nowoursynowska St. 159, 02-787 Warsaw, Poland
4    Faculty of Wood Sciences and Technology, Technical University in Zvolen, 960 01 Zvolen, Slovakia
5    Latvian State Institute of Wood Chemistry, Dzerbenes 27, LV-1006 Riga, Latvia
*    Correspondence: anita_wronka@sggw.edu.pl

**Abstract:** The valorization of tree bark through chemical treatment into valuable products, such as bark acid, leads to the formation of process residues with a high solids content. Since they are of natural origin and are able to be suspended in water and acid, research was carried out on the recycling of suberic acid residues (SAR) as a bi-functional component of binder mixtures in the production of plywood. The 5%–20% (5%–30% for curing time) mass content of SAR has been investigated with urea-formaldehyde (UF) resin of about 66% of dry content. The results show that the curing time of the bonding mixture can be reduced to about 38% and 10%, respectively, for hot and cold curing, of the initial curing time for the lowest SAR content. The decreasing curing time of the tested binder mixtures with the increase in SAR content was caused by the increasing amount of acidic filler, since amine resins as UF require acidification hardening, and the curing dynamics are strongly dependent, among others, on the content of the acid medium (curing agent). In the case of hot curing, a SAR content of about 20% allowed us to achieve the curing time of bonding mass with an industrial hardener. Investigations into the mechanical properties of examined panels showed a significant modulus of elasticity (MOE) increase with filler content increase. Similar conclusions can be drawn when analyzing the results of the modulus of rupture (MOR) investigations; however, these were only significant regarding hot-pressed samples. The shear strength of the plywood samples increased with the SAR rise for both cold- and hot-pressed panels. The in-wood damage of samples with SAR filler, hot-pressed, rose up to about 30% for the highest SAR filler content. For cold-pressed samples, no in-wood damage was found. The positive effect of veneer impregnation limiter by resin was identified for SAR acting as a filler. Moreover, a higher density of SAR-containing bonding lines was reached for hot-pressed panels. Therefore, the results confirmed the ability to use the SAR as an upcycled component of the bonding mixture for plywood production.

**Keywords:** plywood; binder; filler; hardener; bark; suberinic acid; residue; upcycling

## 1. Introduction

Increasingly, attention is being turned to wood waste generated during the harvesting process—branches and bark. Very often these by-products are used as fuel in the incinerators situated next to the factories, or some of these raw materials remain in the forest. Shredded branches were used as an admixture for particleboard [1] or as a face layer of a wood-based composite formed from small slices of branches [2]. When looking for examples of uses for bark, the common use of bark in gardens as mulch is an obvious example [3] but it is also used in the wood-based composites industry, e.g., as an ingredient

in insulation panels [4,5] or as an admixture in the production of particleboard [6,7] due to lower strength parameters [8]. For this reason, research has been conducted for several years to extract selected compounds from the bark–tannins [9]. The aforementioned compounds are currently used in the production of natural wood adhesives [10]. An accessible and ecological binder produced from birch outer bark is hydrophobic suberin, which is usually isolated as suberinic acids or partly depolymerized suberin [11].

To approach the topic comprehensively, attention must be paid to the waste associated with tannin production, which is why, in parallel with the development of an ideal tannin-based wood binder, research is being conducted into the use of SAR for, among other things, bonding particleboard [12,13] and plywood [14]. It has been proven that suberinic-acid-based adhesives can cure without any additives or modifiers, remaining a fully ecological product [15], and at the same time resulting in low thickness swelling and high bending strength [16].

The use of fully biodegradable adhesives, e.g., based on gelatin, milk, and casein, is characterized by lower strength parameters and lower moisture resistance and adhesion to wood [17]; for these reasons, high expectations are placed on adhesives based on biopolymers, e.g., polylactide (PLA) [18]; polycaprolactone (PCL); polypropylene (PP) [19]; and, as previously mentioned, the bark [10] and tannins [20]. The ecological aspect is also very important because the emissions of NOx are increased when wood composites with nitrogen-based adhesives, such as urea-formaldehyde (UF), melamine-urea-formaldehyde (MUF), and emulsion polymer isocyanate (EPI), are combusted [21]. As proven, the durability of already mentioned PP, as well as modified polypropylene glues, is equal to that of UF and MUF resins [22].

One of the ingredients in the glue used to make plywood is filler. It has a crucial role as it prevents the glue from soaking into the veneers too much, and thus reduces its viscosity and reduces material costs but can also be used to modify selected properties or improve electrical/thermal conductivity [23]. We distinguish between organic and synthetic fillers [23], which are divided into active and passive fillers. Organic flour is commonly used as a filler. In the research carried out, it was examined whether the type of flour made a difference in the quality of bonding, for which purpose different types of flour were compared: rye flour, hemp flour, coconut flour, rice flour, and pumpkin flour. The tests confirmed that all the flours used could be successfully used as a filler. Against the background of the others, pumpkin flour proved to be distinctive, whereby the adhesive bond strength was the highest. Hemp flour, on the other hand, had a reducing effect on formaldehyde emissions [24], and pine bark had similar properties [25], as well as beech bark [26]. However, it should be pointed out that the main shortcoming of mentioned flour-based fillers is their food origin. Other examples of fillers in plywood and other ligno-cellulosic composites production are soy flour [27], coffee bean post-extraction residues [28], palm kernel meal [29], oak bark powder [30], thymus plant [31], chestnut shell and coffee waste [32], walnut shell [33], nanocrystalline cellulose [34], and beech bark [35]. In a study using maple bark meal as a filler in the production of three-layer plywood, it was noted that the reduced pH of the bark can lead to the pre-curing of the adhesive. Waste biomass and biomass residues that are abandoned in nature are promising fillers that can be used to replace flour [36]. This knowledge gives hope for the dual use of bark—as a filler and as a curing agent—which can mean considerable savings in the production process [37]. It is not only the chemical properties of the filler that matter but also physical properties, such as the degree of grinding—research confirms that the more ground the filler is, the better the bond quality [38]. Chicken egg shells were used as a bio filler in the other study as well. Due to their non-combustible properties, they can compete with synthetic fillers, and their advantages also include their low price and eco-friendly nature [39]. The examples shown are for commercially used adhesives, but trials can already be found in the literature where fillers are combined with polymers—in this case, wood plastic composites [40]. Starch can also be considered a natural filler used in plywood technology. When combined in the right proportion with citric acid, it produces a biodegradable adhesive bond [41]. It is also used as

a binder in wet-molded fiberboard technology [42]. Wood bark has also found its way into plywood technology so far, including the production of plywood without the use of glue as a binder. The role of a binder was played by oxidized bark (*Acacia mangium*). The powdered bark was oxidized using hydrogen peroxide in four variants: 5, 10, 15, and 20%, as well as a catalyst. The best results were obtained for the 20% variant, and the topic was considered forward-looking and developmental [43]. In conclusion, tannin-based adhesives have great potential on the wood market [44], as evidenced by numerous examples of applications, but the by-products must not be forgotten, since they must also be utilized.

The variety of fillers used confirms the potential for substitutes for wheat or rye flour. The bark post-treatment raw material, prepared as flour/powder, can be successfully used as a filler, so this research will test the use of SAR in plywood-production technology.

This research aimed to evaluate the influence of the application of SAR as a bi-functional component, hardener, and filler in the bonding mixture for plywood production. Such a double function of SAR (filler and hardener), as a natural origin, is a new approach in the research of plywood bonding mass fillers. In the scope of research, the curing time of the bonding mixture with various contents of SAR was measured in a room and at elevated temperatures.

## 2. Materials and Methods

### 2.1. Materials

The rotary cut birch (*Betula* L.) veneer of an average thickness of 1.5 mm, $5\% \pm 1\%$ moisture content (MC), and dimensions of $360 \times 360$ mm$^2$ were used to produce plywood. As a binder, an industrial UF resin S-120 (Silekol Sp. z o.o., Kędzierzyn—Koźle, Poland) of about 66% of dry content [45] was used with ammonium nitrate water solution (industrial hardener mentioned in Table 1; 20 wt%) as a hardener to reach the curing time of REF 0 (Table 1) gluing mass at 100 °C in about 86 s. The rye flour was used as a reference filler. The mentioned UF resin was also the base of the SAR bonding mixture.

**Table 1.** Compositions of bonding mixtures and plywood-selected pressing parameters.

| Variant Label | Filler | Filler Content | Industrial Hardener Content | Pressing Temp. [°C] | Pressing Time [Min] |
|---|---|---|---|---|---|
| | | [pbw [1] per 100 pbw of Solid Resin] | | | |
| REF 0 | Rye flour | 0 | 2 | 120 | 4 min |
| REF 5 | Rye flour | 5 | 2 | 120 | 4 min |
| REF 10 | Rye flour | 10 | 2 | 120 | 4 min |
| REF 20 | Rye flour | 20 | 2 | 120 | 4 min |
| SAR 5 | SAR | 5 | 0 | 120 | 12 min |
| SAR 10 | SAR | 10 | 0 | 120 | 8 min |
| SAR 20 | SAR | 20 | 0 | 120 | 4 min |
| SAR 5 C | SAR | 5 | 0 | Room | 24 h |
| SAR 10 C | SAR | 10 | 0 | Room | 24 h |
| SAR 20 C | SAR | 20 | 0 | Room | 24 h |

[1] pbw—parts by weight.

The SAR were used in this research as an alternative filler and hardener. SAR was kindly provided by the Latvian State Institute of Wood Chemistry, Riga, Latvia, made of the residues resulting from the isolation of suberinic acid in the ethanol, in the process described by [12], suspended in water and acidified to pH 2, filtered and rinsed with deionized water, which is described by [13]. The SAR, delivered as a brown paste (Figure 1a), about 25% of dry matter, pH 3.3, was dried at 70 °C to constant mass. The chemical properties are described in [12]—acid number 95.8 mg/KOH, epoxy groups 0.61 mmol/g, cellulose 9.0 wt%, aromatic suberin, lignin 21.4 wt%, ω-hydroxy acids 17.5%, and α, ω-diacids 11.9%. Then, the totally dry matter was crushed in an SM100 cutting mill (RETSCH GmbH,

Haan, Germany), and the fine fraction <0.1 mm (extracted by sieving) was taken for further research (Figure 1b). The pH of the water solution of the achieved SAR powder was 3.7.

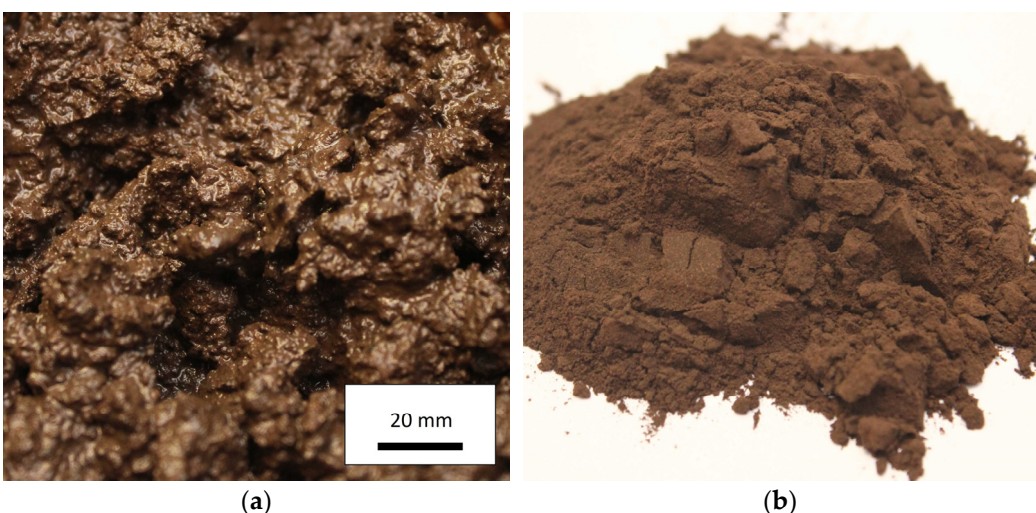

**(a)** **(b)**

**Figure 1.** The SAR in the wet state (as delivered) (**a**) and dry state (**b**).

*2.2. Methods*

2.2.1. Bonding Mixtures Curing Time

To measure the curing time of the bonding mixtures of various contents of different fillers, the mixtures of binders were prepared, as given in Table 1. Additionally, the mixture of UF resin and SAR filler in the ratio of 100:30 pbw was prepared. This variant of bonding mixture was not used for further plywood preparation due to extremely high viscosity. The samples of mixtures of a weight of about 8 g each were moved to laboratory glass tubes, and the time to cure was measured during storage of samples in glass tubes in: (1) boiling water (hereinafter called "Hot") or (2) room temperature (about 20 °C; hereinafter called "Cold"; SAR samples only) accompanied by manual slow stirring. As many as 5 repetitions of curing time measurement were performed for every bonding mixture and every temperature mentioned.

2.2.2. Plywood Preparation and Characterization

The three-layer plywood sheets were prepared with use of the bonding mixtures listed in Table 1. The following parameters were applied during plywood preparation: bonding mixture spread 170 g m$^{-2}$ (manually spread) and maximum press pressure 1.4 MPa for hydraulic press (AKE, Mariannelund, Sweden). The pressing time is given in Table 1. All the produced plywood sheets were subjected to conditioning at 20 °C/65% ± 1% R.H. to constant weight before further testing.

The following tests were completed for the produced plywood: bonding quality in a dry state according to [46] (8 repetitions), modulus of elasticity (MOE), and modulus of rupture (MOR) in a parallel direction to the grains of the face veneer layer (8 repetitions), according to [47]. All the mechanical tests were performed on a computer-controlled universal testing machine (Research and Development Centre for Wood-Based Panels Sp. z o.o. Czarna Woda, Poland). The density profile of the tested plywood was measured (3 repetitions) on a GreCon DAX 5000 device (Fagus-GreCon Greten GmbH and Co. KG, Alfeld/Hannover, Germany) with sampling step of 0.02 mm. The density profile measurement results were the representative plots selected after analyses of 3 individual plots for every tested panel.

The images of the cross-cuts of the investigated plywood samples were taken with a Nikon SMZ 1500 (Kabushiki-gaisha Nikon, Minato, Tokyo, Japan) optical microscope. The images of the break zone of the samples after the shear strength test (bonding quality test)

were taken with the use of a Canon 550D digital camera (Canon Inc., Tokyo, Japan). The in-wood damage (fiber failure) was estimated, according to [46].

### 2.3. Statistical Analysis

Analysis of variance (ANOVA) and t-tests calculations were used to a test ($\alpha = 0.05$) for significant differences between factors and levels, where appropriate, using IBM SPSS statistic base (IBM, SPSS 20, Armonk, NY, USA). A comparison of the means was performed when the ANOVA indicated a significant difference by employing the Duncan test. The statistically significant differences for the achieved results are given in the Results and Discussion paragraphs when the data are evaluated. The letters on the plots indicate the homogeneous groups; however, the homogeneity has been tested among the three groups (REF, SAR hot, and SAR cold) in different filler content and curing temperature.

## 3. Results and Discussion

### 3.1. Curing Time

The results of the curing time measurement of bonding mixtures of various content and different types of filler are presented in Figure 2. As is shown in the case of hot curing, the curing time rapidly decreases from 225 s for SAR content 5 pbw to 86 s for SAR content 30 pbw. A similar tendency has been found in the case of cold curing, where the curing time was decreasing from 1260 min for SAR content 5 pbw to 120 min for SAR content 30 pbw. All the analyzed average values have been statistically significantly different from one to another. The curing time of the reference bonding mixture (REF 0) was 86 s. The decreasing curing time of the tested bonding mixtures with increasing SAR content was caused by the rising amount of acidic-nature filler. According to [48], the amine resins, including UF, need to be acidified to be cured, and the curing dynamic is strongly dependent, i.a., on the content of the acidic medium (hardener). However, it should be pointed out that the decreasing time has an asymptotic character, which means the intensity of the curing time decrease is lower for higher filler content than for lower filler content. This is also influenced by the phenomena of buffering capacity [49]. The further addition of acidic filler will not affect curing time shortening, as it was found for lower filler content.

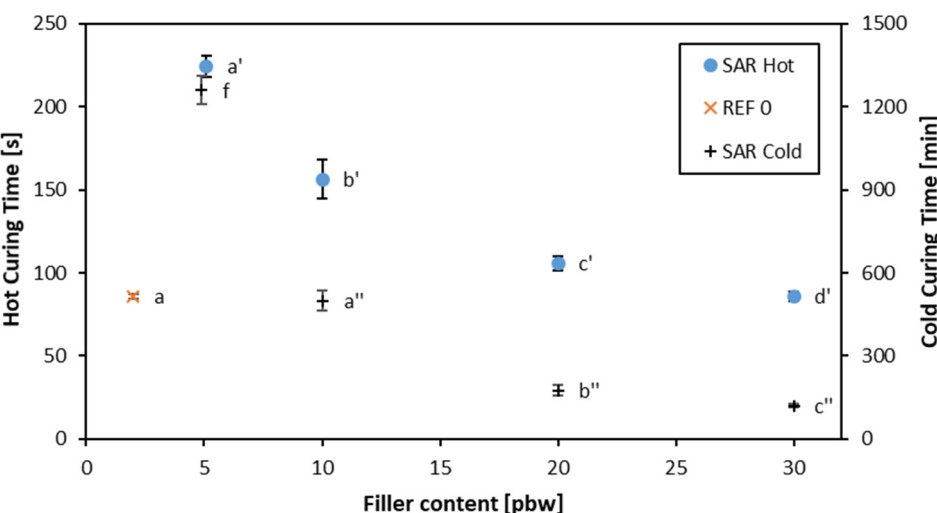

**Figure 2.** The dependence of the curing time vs. filler content (the letters on the plots indicate the homogeneous groups).

### 3.2. Bonding Quality

The following features of the produced plywood sheets were investigated to estimate the quality of bonding: shear strength and images of the break zone of the samples after the shear strength test. The results of the shear strength measurement are presented in Figure 3, and the images of the break zone are provided in the pictures collected in Figure 4. As is

presented in Figure 3, for all the tested bonding mixtures used, the shear strength increased linearly with the filler content increase. The most intensive increase in shear strength has been found for the reference bonding mixture with rye flour as a filler, where the shear strength increased from 1.06 N mm$^{-2}$ for 0 filler content to 1.71 N mm$^{-2}$ for 20 pbw filler content. The increase in SAR filler content from 5 pbw to 20 pbw caused shear strength to increase from 1.43 N mm$^{-2}$ to 1.57 N mm$^{-2}$. That means, the SAR content increase did not provide as much dynamic shear strength improvement as it did for the reference bonding mixture. However, it is worth mentioning that for a similar filler content, 5 pbw, better shear strength has been found for SAR filler (SAR hot samples). The SAR filler content increase for the bonding mixture cured at room temperature led to a slight shear strength increase from 0.25 N mm$^{-2}$ to 0.42 N mm$^{-2}$ for the SAR 5 C and SAR 20 C bonding mixtures, respectively. Regarding the statistical significance of differences in average values of shear strength, in the case of the reference binder, the REF 0 was significantly different from the remaining REF panels, as well as REF 20 being significantly different from the remaining samples. The only statistically significant difference in average shear strength for both SAR hot and SAR cold variants was found for SAR 20 compared to the remaining samples.

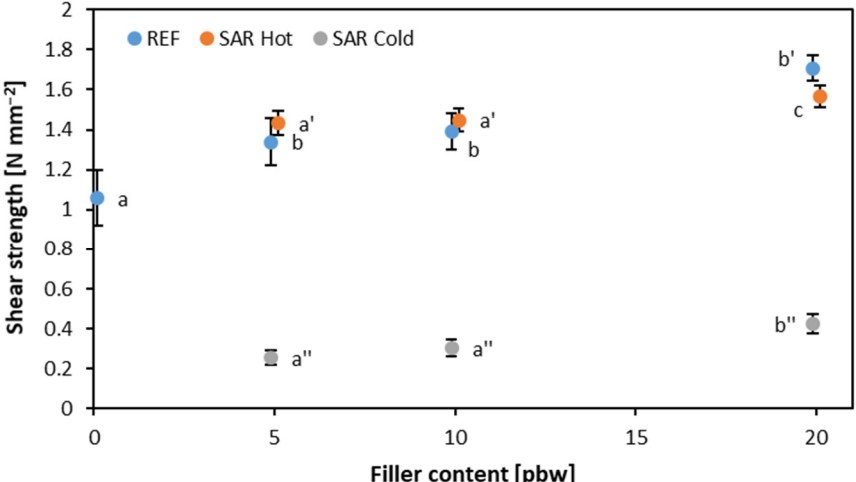

**Figure 3.** The shear strength of the plywood samples of different filler content (the letters on the plots indicate the homogeneous groups).

Interesting observations can be found when analyzing the images of the break zones of the samples after the shear strength test (Figure 4). For reference panels (REF 0, REF 5, REF 10, and REF 20), the increasing filler content led to the rise of the in-wood destruction area. For SAR hot samples (SAR 5, SAR 10, and SAR 20), the same tendency can be observed. For cold-pressed panels (SAR 5 C, SAR 10 C, and SAR 20 C), only the rising darkness intensity of the bonding line has been found due to increasing SAR filler content without the change in the in-wood damage. As confirmed by [17,50], the in-wood damage area was strongly connected to the shear strength of the bonded wood samples. The significant influence of the bark filler content on the bonding shear strength has also been confirmed by [37,51,52].

The differences in the break zones of the plywood samples bonded with different contents and types of filler pressed in the hot or cold press process after the shear strength test are collected in the pictures that are presented in Figure 4. According to [53,54], the analyses of layered materials break zone help with the evaluation of the properties of materials and allow for the description of the relations between the production parameters and material features. In the case of reference samples (REF 0, REF 5, REF 10, and REF 20), with the increasing filler content, the in-wood damage of the connection significantly rises, especially in the REF 20 sample. There was no in-wood damage for REF 0 and less than 10% for REF 5 and REF 10, while there was about 30% for REF 20. The in-wood damage of samples with SAR filler, hot-pressed, rose from 0 for SAR 5, through 10% for SAR 10

to about 30% for SAR 20 (lowest to highest SAR filler content). For cold-pressed samples (SAR 5 C, SAR 10 C, and SAR 20 C), except for the progressively darker surface, caused by an increasing content of the brown filler, no in-wood damage has been found.

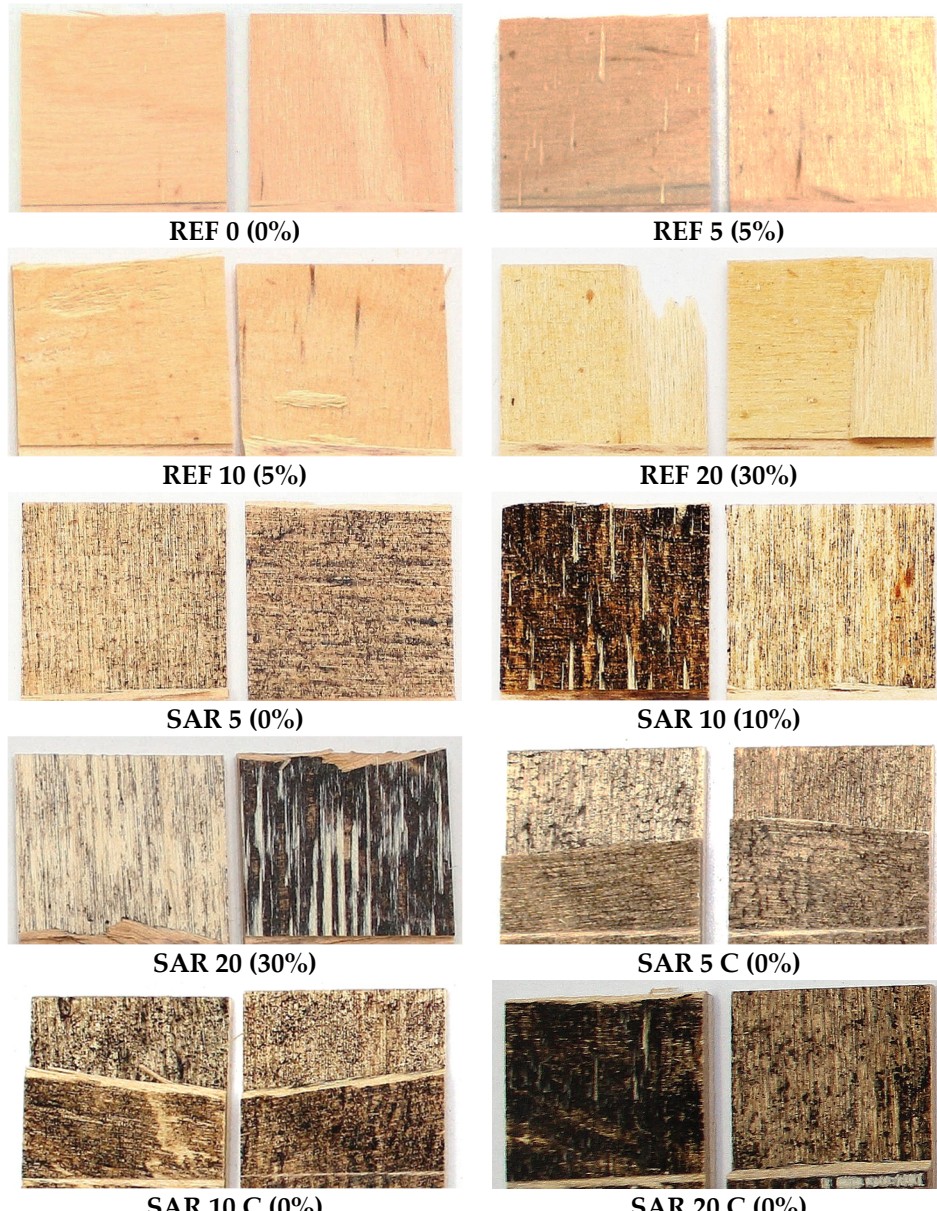

**Figure 4.** The images of the break zones of the samples after the shear strength test (dimensions: $25 \times 25$ mm$^2$; numbers in brackets are the estimation of in-wood damage).

### 3.3. Modulus of Elasticity and Modulus of Rupture

The modulus of elasticity results are presented in Figure 5. A significant MOE increase has been found with filler content increase; however, the maximum MOE has been noted for about 10% of filler content. Further rise of filler content reduces MOE. The difference between MOE for 10% filler content compared to the remaining samples of reference binder were statistically significant. According to the state of the art [35], the filler addition of more than 15% led to a radical binder viscosity increase. As was stated for bark-based filler for plywood bonding mass [55], the increase in filler content of about 20% significantly increases the viscosity of the binder. Thus, the possible problems with the even spreading of the binder over the veneer surface can negatively influence the bending strength of the plywood sheets. When analyzing the results for SAR hot-pressed panels, it can be

concluded that within the tested filler content of 5%–20%, the filler content increase caused the rise in MOE. What is more, all the achieved results are statistically significantly different from one to another, and high repeatability of the MOE results has been found (low standard deviation values indicated on the plots as the error bars). The increasing tendency of MOE for SAR cold-pressed panels has also been found, and it can also be seen that all the achieved results are statistically significantly different from one to another.

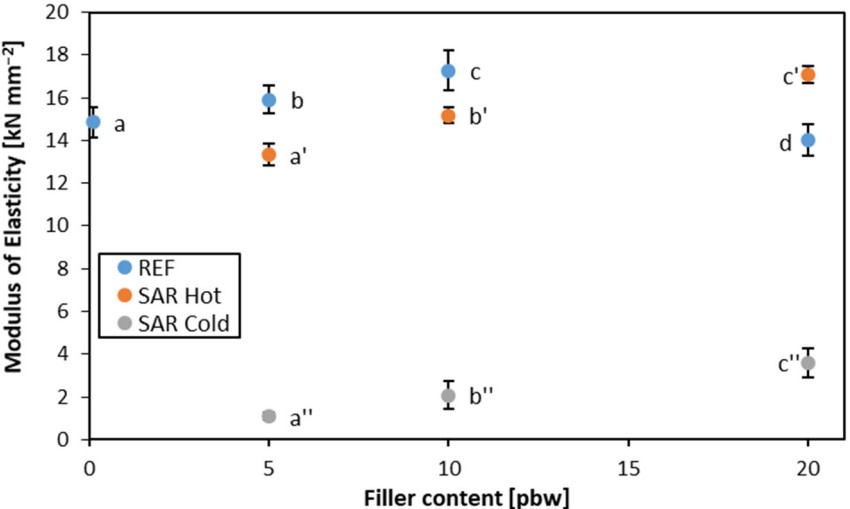

**Figure 5.** The modulus of elasticity of the plywood samples of different filler content (the letters on the plots indicate the homogeneous groups).

Similar conclusions can be drawn when analyzing the results of MOR investigations (Figure 6). However, in the case of cold-pressed panels, no significant influence of SAR filler content has been found. Again, for industrial filler, the optimal content, leading to maximum MOR, was about 10%. This is in line with the findings of [29], where the filler content in the range of 10%–15% was indicated as optimal for the mechanical properties of the layered wood composite.

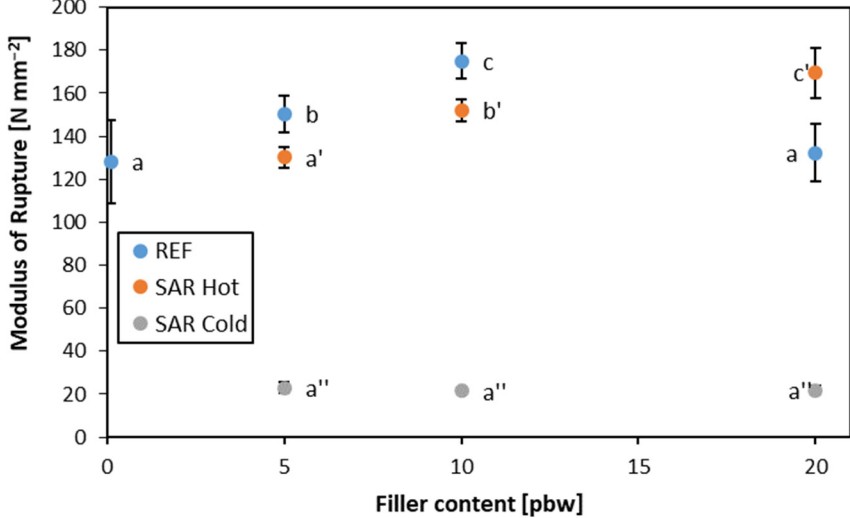

**Figure 6.** The modulus of rupture of the plywood samples of different filler content (the letters on the plots indicate the homogeneous groups).

The lower content of the filler decreases MOE and MOR due to the insufficient bonding line creation, caused by too high of an impregnation of the veneer by the bonding mixture. Thus, the break of the bent sample occurs in the bonding line (between veneers), as is

shown in Figure 7. In the same figure, the destruction of the bent sample by the break of the bottom layer has been shown, where the highest values of MOE and MOR were reached.

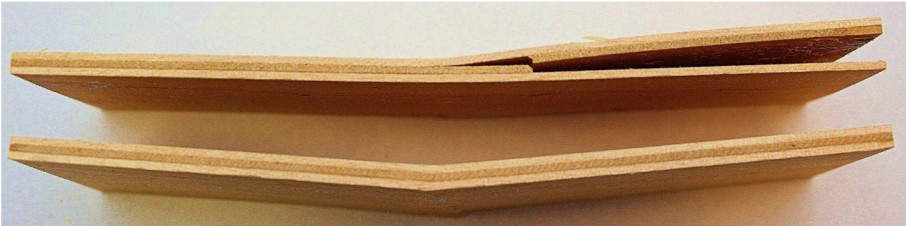

**Figure 7.** The image of samples after bending test: REF 0 delamination in bonding line (**top**) and REF 10 bottom layer break (**bottom**).

*3.4. Density Profile*

Figure 8 shows the course of density profiles for control samples, without SAR. The shape of the density profiles is generally symmetrical. The density of the veneers differed between the variants, ranging from 600 kg m$^{-3}$ to 800 kg m$^{-3}$, without a special relation to production parameters. The average thickness of the samples was about 4.4 mm. The bonding lines showed a significant increase in density across the sample, reaching 1000 kg m$^{-3}$ for REF 0 (without filler addition) and about 1100 kg m$^{-3}$ for the other variants. One can see here that the relationship between the density of the binder and the mass fraction of the filler—the higher its share, the greater the density of the bonding line.

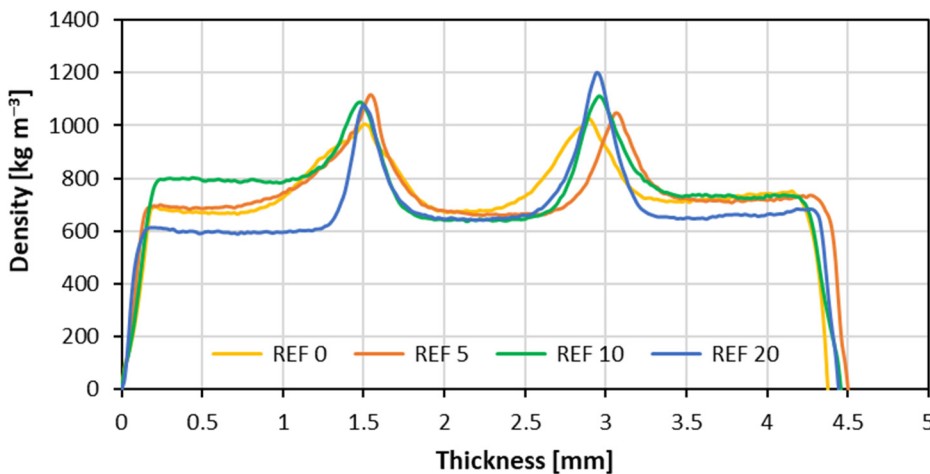

**Figure 8.** The density profiles of the reference plywood samples of different filler content.

The density profiles of the hot-pressed samples prepared with SAR as the filler (Figure 9) showed a symmetrical course. The density of the veneers was, depending on the sample, 700–800 kg m$^{-3}$, while the density of the bonding line ranged from about 1100 kg m$^{-3}$ for samples with a mass fraction of SAR of 5 to about 1300 kg m$^{-3}$ for samples with a mass fraction of SAR of 20. That means an increase in bonding line maximum density has been found with a SAR filler content increase. A slight rise in the panels' densification can be found, since the mean thickness of the samples was about 4.3 mm.

In Figure 10, the density profiles of cold-pressed plywood with SAR as the filler have been presented. The density profile of three-layer cold-pressed plywood is symmetrical for all three variants. The veneer layers showed a density of about 600 kg m$^{-3}$. For samples with the share of SAR as a filler of 20 parts by mass, the adhesive joint showed a higher density of about 900 kg m$^{-3}$. However, for the remaining variants—5 and 10 mass parts of SAR—the test showed a decrease in density at the point of the adhesive joint, most clearly visible for the smallest share of SAR—about 600 kg m$^{-3}$, below the density values achieved by the veneer. Between the decrease in density and the veneer layers, there was a

brief increase in density—presumably at the point of contact between the wood and the glue, i.e., in the layer of wood where the glue soaked. A smaller decrease in density is also seen for samples with a mass fraction of SAR of 10—here, after an increase in the value to over 820 kg m$^{-3}$, a decrease to lower values, closer to 800 kg m$^{-3}$, can be observed. The described observations are confirmed by the research of [56], which proved that in the case of layered materials, the glue line bonds for all the dry-pressed plies (glue-formulated and control) are stronger than those for the cold-pressed samples before and after field exposure.

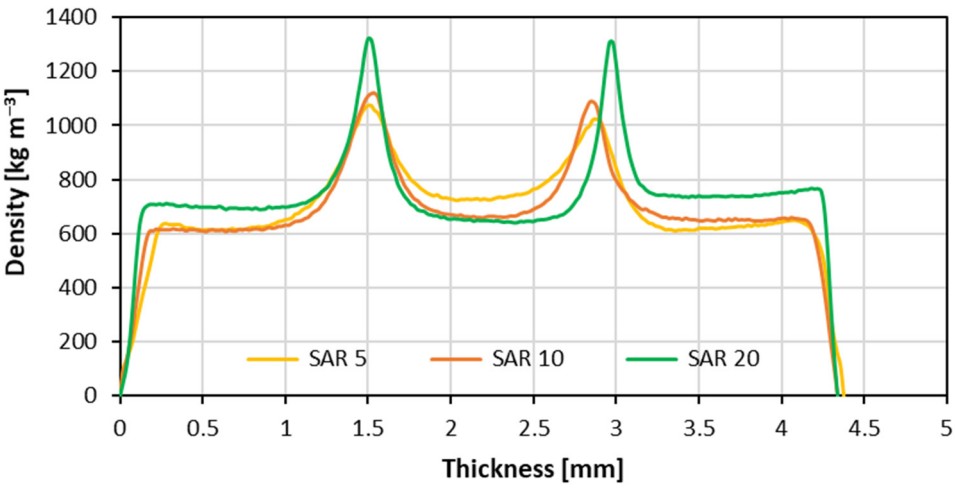

**Figure 9.** The density profiles of the plywood samples of different SAR filler content, hot-pressed.

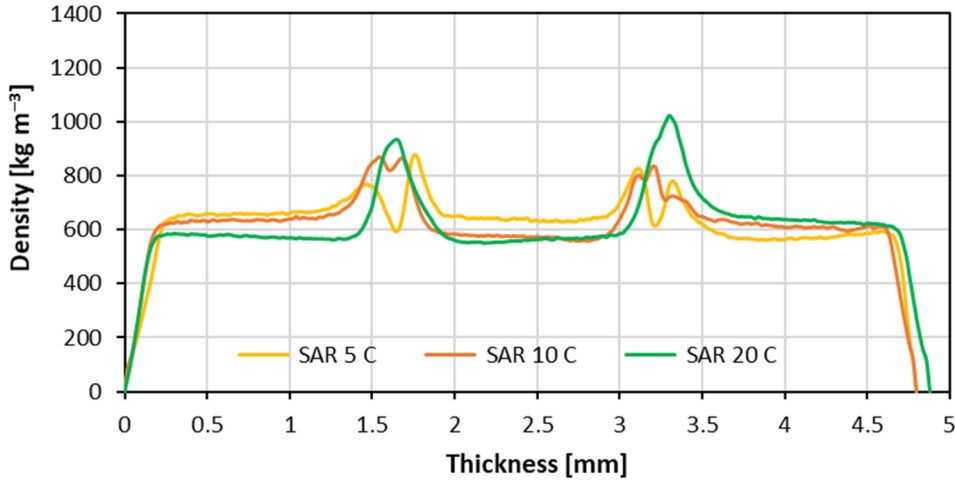

**Figure 10.** The density profiles of the plywood samples of different SAR filler content, cold-pressed.

What is more, since the cured resin has a density higher than wood, the density profiles allow estimations of the depth of wood penetration by resin. As can be seen in Figure 8, the penetration depth has been the highest for REF 0 samples, which can be explained by the lack of filler. As has been confirmed by [29], the rise in filler content significantly reduces the depth of soaking of resin by veneers. In the case of REF 0 panels, the depth of penetration was about 0.7 mm. With raising filler content, the penetration depth decreases to about 0.3 mm. In the case of the bonding mixture with SAR filler and hot-pressed panels, the depth of penetration of veneers by resin decreased from 0.5 mm for SAR 5 (lowest filler content) to about 0.3 mm (similar to industrial filler) for SAR 20 (highest filler content). The specific relation between the bonding mixture and veneer has been found for the bonding mixture with SAR filler and cold-pressed panels. The potential depth of penetration should be investigated along with the analysis of the cross-cut of samples. Due to the relatively high viscosity of the bonding mixtures with fillers, especially those of bark origin, the

viscosity of the bonding mixture is high and rises over time [24]. The hot pressing, due to rising temperature, helps to decrease the viscosity of that mixture [29]. Thanks to this, the bonding mixture of lower viscosity can penetrate deeper into the wood structure. That was the case with hot-pressed samples. This way, better adhesion can be achieved, where the adsorption phenomenon is strongly supported by mechanical anchorage [57]. When cold-pressing is conducted, there is no factor leading to lower viscosity. This is why the penetration of the bonding mixture of cold-pressed panels was extremely low. This can be visible in Figure 11, where the comparison of the bonding lines is shown. It is hard to estimate the thickness of the bonding line for the REF 10 sample when the bonding line thickness for SAR 10 sample (hot-pressed) was almost 0, but the resin penetration is well visible and is about 0.4 mm, which is in line with the findings that came out from the Figure 9 analysis. In the case of the SAR 10 C sample presented in Figure 11, there is no penetrated (dark) zone in the wood structure. In that case, the thickness of the bonding line is about 0.1–0.15 mm. According to [29], the increasing content of the filler in the plywood bonding mixture leads to an increase in bonding line thickness. Here, the only adhesion interface is based on the adsorption phenomenon, with weak mechanical anchorage due to low penetration.

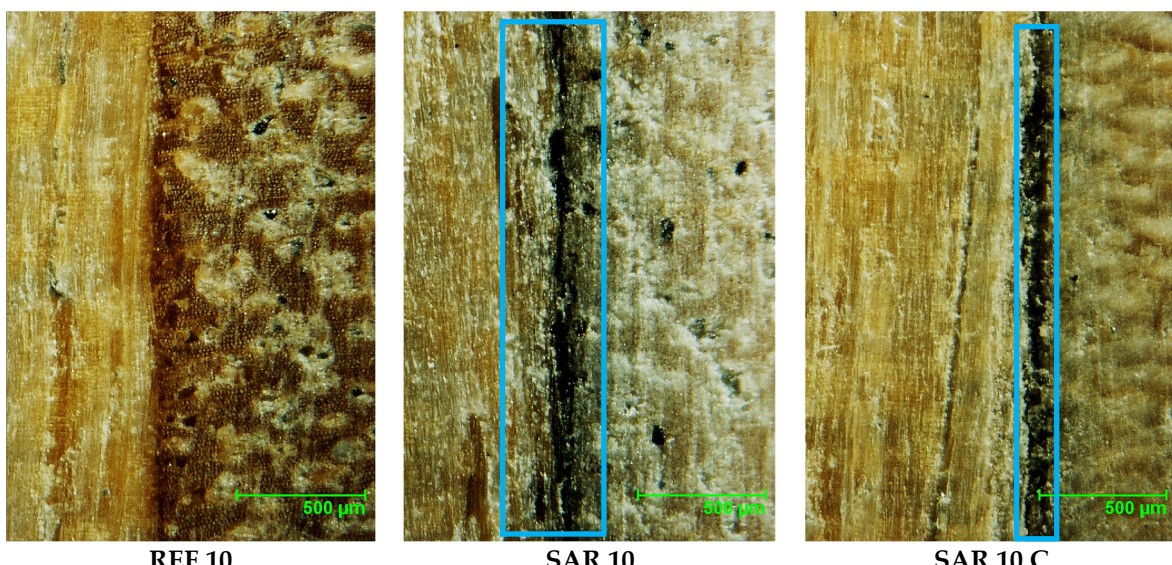

| REF 10 | SAR 10 | SAR 10 C |

**Figure 11.** The images of the plywood samples bonding line of the same filler content (vertical bonding line in the middle of pictures; depth of binder penetration highlighted by blue frames).

## 4. Conclusions

In this research, an attempt has been made to upcycle the bark post-treatment residues that came from suberinic acid production to a bi-functional component of bonding mixture in plywood production. The acid-character SAR acts as a filler and hardener of a bonding mixture based on urea-formaldehyde resin. The 5%–20% (5%–30% for curing time) mass content of SAR has been investigated. The activity of SAR filler has been tested in a room at elevated temperatures.

The results show that in the case of hot-curing, the curing time of the bonding mixture can be reduced to about 38% of the initial curing time for the lowest SAR content. The curing time of samples cured at room temperature has been reduced to less than 10% of the initial curing time. In the case of hot curing, the SAR content of about 20% allows one to achieve the curing time of bonding mass with an industrial hardener.

The shear strength of the plywood samples increases with the SAR rise for both cold- and hot-pressed panels. The positive effect of veneer impregnation limiter by resin has been identified for SAR acting as a filler. Additionally, a higher density of SAR-containing bonding lines has been reached for hot-pressed panels. In the case of bending, strength,

and modulus of elasticity, the increase in both parameters has been found when increasing the SAR filler content within the above-mentioned range.

The results confirmed the ability to use the SAR as an upcycled component of the bonding mixture for plywood production. Further, the details concerning binder viscosity tuning by either temperature and/or water addition should be investigated to improve the modified binder features.

**Author Contributions:** Conceptualization, A.J., A.W., L.K., J.R. and G.K.; Data curation, A.W. and G.K.; Formal analysis, L.K., R.R., J.R. and G.K.; Funding acquisition, J.R. and G.K.; Investigation, A.J., A.W., A.D. and G.K.; Methodology, A.J., L.K., J.R. and G.K.; Project administration, A.W. and G.K.; Resources, A.W., J.R. and G.K.; Supervision, G.K.; Validation, L.K. and R.R.; Visualization, A.J. and A.D.; Writing—original draft, A.J., A.W. and G.K.; Writing—review and editing, A.J., A.W. and G.K. All authors have read and agreed to the published version of the manuscript.

**Funding:** This research has been funded by the National Science Centre, Poland, under the Forest-Value 2021 Programme, reg. no. 2021/03/Y/NZ9/00038, project acronym BarkBuild, as well as being supported by the Slovak Research and Development Agency under contracts APVV-19-0269 and No. SK-CZ-RD-21-0100.

**Data Availability Statement:** Not applicable.

**Acknowledgments:** The mentioned research has been completed with the support of the Student Furniture Scientific Group (Koło Naukowe Meblarstwa), Faculty of Wood Technology, Warsaw University of Life Sciences—SGGW.

**Conflicts of Interest:** The authors declare no conflict of interest.

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
