# Peer review of "Influence of Upcycled Post-Treatment Bark Biomass Addition to the Binder on Produced Plywood Properties"

_forests, doi:10.3390/f14010110_

Round 1

Reviewer 1 Report

The manuscript ‘Influence of upcycled post-treatment bark biomass addition to the binder on produced plywood properties’ submitted for the probable publication in the journal Forests.

This manuscript deals with the suitability of using of suberinic acid residues (SAR) as a bi-functional component of bonding mixtures in plywood production. Experimental showed that curing time can be reduced and bending properties of plywood can be improved by using increasing amount to of SAR. This research work is well designed, and manuscript is well-written with consistency. There is novelty in this work and research community can get benefit from the outcome of this research work. However, I have some comments and those need to be addressed before the accepting this manuscript.

Line 245: It would be a figure, not a table. I recommend to measure the fiber failure according to the standard EN 314-1. It would provide much information about apparent cohesive wood failure.

Line 332-333: How you came to on the conclusion ? Have you measured the penetration depth of the glue? There is a specific method for measuring effective and mean penetration.

Line 340-341: This sentence is difficult to understand. Please rewrite it.

Figures 2-5: Please provide bar graphs for better comparisons. Linear regression analyses are completely unnecessary. Please provided bar graphs with error bars and lettering to see the significant differences as you have performed Duncan multiple range tests.

Figure 10. Better visualization of bondline can be seen from the cross section and samples must be prepared with extra care. Optical image is not good enough. SEM pictures are needed for better visualization.

Reviewer 2 Report

Original and well-structured paper. His reading is clear and concise.

The methodology is adequate and well developed

The results are clear and the figures are all necessary.

In table 2, in each photo, I would indicate some issues with arrows to highlight where the wood breaks and what is the presence of adhesive. In figure 10, I would indicate with arrows the penetration of the adhesive.

Reviewer 3 Report

Dear Editor

The manuscript entitled “Influence of upcycled post-treatment bark biomass addition 2 to the binder on produced plywood properties”, main concerns include:

1.     The Abstract is so primitive, rewrite it.

2.     Interfical adhesion and chemical analysis are mandatory.

3.     What about the aesthetic cost of bark addition?

4.     The novelty should be highlighted as there are a lot of already published papers in this field.

5.     Some references do not back the claims and justifications needed.

6.     The introduction lacks a strong background study. The potential findings of other potential research must be included in the background. Authors should use and cite other important publications that used natural extracted as a filer/extender and explain their shortcomings that encouraged authors to work on another approach. The last section of the introduction must clearly focus on objectives, a summary of the methodology, and a contribution to the scientific society. Additionally, highlight the novelty and scope in the introduction.

7.     In Fig 6. No sign of bark addition is evident.

8.     What about the resulting resin and or any potential treatment for it?

9.     Following literature can highly improve this manuscript;

https://doi.org/10.1016/j.matdes.2013.05.007

https://doi.org/10.1016/j.compositesb.2014.11.009

https://doi.org/10.1016/j.eaef.2014.07.003

Long term hygroscopic characteristics of polypropylene based hybrid composites with and without organo-modified clay | SpringerLink

Influence of walnut shell as filler on mechanical and physical properties of MDF improved by nano-SiO2 | SpringerLink

Supplementation of Natural Tannins as an Alternative to Formaldehyde in Urea and Melamine Formaldehyde Resins used in MDF Production

The effect of resin type and strand thickness on applied properties of poplar parallel strand lumber made from underutilized species | SpringerLink

I would like to review the revised paper.

Round 2

Reviewer 1 Report

Thanks for correcting your manuscript according to the comments. Still figures 2,3,5 and 6 need more description in their captions. Please indicate you have tested homogeneity among the three groups (REF, SAR Hot and SAR cold) in different filler content.  Use lettering like a, a' and a'' in three groups for better comparison.

Line 215 has word in other language. 

Reviewer 3 Report

The manuscript significantly improved, and it can be accepted for publication.

Author Response

Dear Reviewer, thank you again for your remarks provided previously and for acceptation of our effort to improve the manuscript!